# Akt Is Controlled by Bag5 through a Monoubiquitination to Polyubiquitination Switch

**DOI:** 10.3390/ijms242417531

**Published:** 2023-12-15

**Authors:** Ismael Bracho-Valdés, Rodolfo Daniel Cervantes-Villagrana, Yarely Mabell Beltrán-Navarro, Adán Olguín-Olguín, Estanislao Escobar-Islas, Jorge Carretero-Ortega, J. Alberto Olivares-Reyes, Guadalupe Reyes-Cruz, J. Silvio Gutkind, José Vázquez-Prado

**Affiliations:** 1Department of Pharmacology, Cinvestav-IPN. Av. Instituto Politécnico Nacional 2508, Col. San Pedro Zacatenco, Mexico City 07360, Mexico; ismael.bracho@edu.uag.mx (I.B.-V.);; 2Academic Department of Apparatus and Systems I, Deanship of Health Sciences, Universidad Autónoma de Guadalajara, Av. Patria 1201, Zapopan 45129, Mexico; 3Department of Pharmacology, Moores Cancer Center, School of Medicine, University of California San Diego, La Jolla, San Diego, CA 92093, USA; 4Department of Biochemistry, Cinvestav-IPN. Av. Instituto Politécnico Nacional 2508, Col. San Pedro Zacatenco, Mexico City 07360, Mexico; 5Department of Cell Biology, Cinvestav-IPN. Av. Instituto Politécnico Nacional 2508, Col. San Pedro Zacatenco, Mexico City 07360, Mexico

**Keywords:** Akt, BAG-5, Bcl-2 associated athanogene 5, ubiquitin-dependent degradation, Hsp70

## Abstract

The serine-threonine kinase Akt plays a fundamental role in cell survival, metabolism, proliferation, and migration. To keep these essential processes under control, Akt activity and stability must be tightly regulated; otherwise, life-threatening conditions might prevail. Although it is well understood that phosphorylation regulates Akt activity, much remains to be known about how its stability is maintained. Here, we characterize BAG5, a chaperone regulator, as a novel Akt-interactor and substrate that attenuates Akt stability together with Hsp70. BAG5 switches monoubiquitination to polyubiquitination of Akt and increases its degradation caused by Hsp90 inhibition and Hsp70 overexpression. Akt interacts with BAG5 at the linker region that joins the first and second BAG domains and phosphorylates the first BAG domain. The Akt-BAG5 complex is formed in serum-starved conditions and dissociates in response to HGF, coincident with BAG5 phosphorylation. BAG5 knockdown attenuated Akt degradation and facilitated its activation, whereas the opposite effect was caused by BAG5 overexpression. Altogether, our results indicate that Akt stability and signaling are dynamically regulated by BAG5, depending on growth factor availability.

## 1. Introduction

The serine/threonine kinase Akt (also known as PKB) plays an integral role in maintaining metabolic homeostasis and cell survival and also contributes to cell proliferation and motility, stimulated by growth factor receptors, G protein-coupled receptors, and adhesion receptors, among others [1,2]. Thus, different signaling inputs control Akt recruitment to the plasma membrane and activation via interactions with 3-phosphorylated phosphoinositides and regulatory proteins, involving its phosphorylation by PDK1 and mTORC2 [3,4,5,6].

Long-term control of Akt occurs in response to conditions that affect its phosphorylation and stability, including drug resistance to cancer precision therapies targeting PI3K [7,8,9,10,11,12,13,14]. For instance, during its synthesis, the stability of Akt is maintained by mTORC2-dependent phosphorylation at Thr-450 [15,16]. In contrast, degradation of Akt follows its maximal activation, also driven by mTORC2, in this case by phosphorylation at Ser473, eventually resulting in ubiquitination-linked degradation [17,18]. In addition, degradation of Akt is caused by adverse conditions such as oxidative stress and growth factor withdrawal [9,19,20,21]. Importantly, activation of Akt is also facilitated by ubiquitination, which contributes to the recruitment of Akt to the plasma membrane as a step forward in the response to growth factor-dependent signaling [10,11,12]. The differential fate of polyubiquitinated proteins depends on the geometry of the polyubiquitin chain, which leads to degradation when the chain is linked to Ub-K48 and controls signaling when linked to Ub-K63. In contrast, monoubiquitination regularly controls protein trafficking [22].

Akt stability and signaling are intrinsically linked to its interaction with molecular chaperones such as Hsp90 and Hsp70. Hsp90 stabilizes Akt, preventing its degradation [15,23], whereas Hsp70 either stabilizes or promotes Akt degradation, depending on the control of Hsp70 ATPase activity [24]. The precise mechanism by which Hsp70 leads to Akt degradation remains unsolved, but it might involve, as we propose here, the participation of interacting proteins such as BAG5 (Bcl-2 associated athanogene 5), which may constitute a dynamic functional bridge between them. BAG5 is a multidomain co-chaperone that interacts with the nucleotide-binding domain of Hsp70, promoting the exchange of ADP for ATP that activates the chaperone function [25]. In addition, BAG5 regulates E3 ubiquitin ligases, controlling protein mono- and polyubiquitination [26,27], and also controls autophagosome-dependent degradation [28]. Here, we tested the hypothesis that BAG5, which we identified as an Akt-interacting protein, directly targets Akt to a degradation system active in response to known Akt-destabilizing conditions.

## 2. Results

### 2.1. Identification of BAG5 as an Akt-Interacting Protein

Akt is a key serine/threonine kinase constituted by a pleckstrin homology domain followed by a central catalytic kinase domain and a regulatory domain extended towards the carboxyl terminus (Figure 1A). In order to identify new potential regulators and substrates with an affinity for the catalytic domain of Akt, we used its kinase domain with a K179M mutation, characteristic of a dominant negative version of this kinase (Akt-KDD) [29], as bait in a yeast two-hybrid assay to screen a commercial human fetal brain cDNA library and cloned full-length BAG5 as an Akt-interacting protein (Figure 1A). In the yeast two-hybrid system, the specificity of the interaction between Akt and BAG5 was demonstrated using the growth of cells in the absence of histidine, attributable to the expression of Gal4-dependent genes that also promoted the expression of an α-galactosidase reporter, only of those cells transformed with the combination of plasmids coding for Akt-KDD and BAG5 or, as positive controls, p53 and the SV40 large T antigen (pTD1), used due to their known interaction (Figure 1B, right panel). Yeast grew in media lacking leucine and tryptophan, which selected for the presence of the different transformed plasmids (regardless of potential protein interactions) (Figure 1B, left panel), further indicating that in media lacking histidine, leucine, and tryptophan (-HLT), the absence of growth in cells transformed with Akt-KDD and pTD1 or BAG5 and p53 (Figure 1B, right panel) was in fact due to the lack of interactions between these proteins. These results confirmed the specificity of the interaction between Akt and BAG5 in the yeast two-hybrid system. To assess whether Akt and BAG5 interact in mammalian cells, the human epithelial cell line HEK293T was transfected with full-length Akt1 and BAG5, tagged with HA in the case of Akt and either Myc or GST in the case of BAG5, and the potential interaction between these proteins was tested using co-immunoprecipitation or pull-down. Consistent with the results of the yeast two-hybrid assay, Myc-BAG5 co-immunoprecipitated with HA-Akt (Figure 1C), whereas HA-Akt was detected in the GST-BAG5 pull-down, but almost nothing in the control pull-down containing an equivalent amount of isolated GST (Figure 1D).

BAG5 belongs to a group of chaperone regulators whose structure is characterized by the presence of at least one BAG domain [30,31]. In this case, BAG5 contains four BAG domains, one at the amino terminus followed by a linker sequence and three additional BAG domains towards the carboxyl terminus (Figure 1A) [30,32]. To identify the BAG5 domain involved in the interaction with Akt, we generated different BAG5 constructs fused to GST, either containing or lacking the linker region (Figure 1E). Pull-down assays revealed the importance of the BAG5 linker region for its interaction with Akt, as revealed by the ability of all GST-BAG5 constructs containing this region, but not GST alone, to interact with Akt (Figure 1F). The BAG5 linker region, extending from N87 to P181, showed a higher level of interaction with Akt, even though this construct was expressed at lower levels compared to other BAG5 constructs (Figure 1F). The interaction was specific, as indicated by the absence of Akt in the pull-down of GST and the lack of significant interaction between Erk and the BAG5 constructs (Figure 1F, upper panel). In these experiments, transfection of HA-Akt and HA-Erk was adjusted to obtain similar levels of expression of both kinases (Figure 1F, lower panels). Moreover, the catalytic domain of Akt (with the mutation K179M equivalent to the bait used in the yeast two-hybrid system), in this case fused to EGFP, specifically interacted with the BAG5 linker region, expressed as a GST-fusion protein, but not with GST used as control, as revealed by its presence only in the lane corresponding to pulled-down GST-BAG5(N87-P181) (Figure 1G, left panel). We next examined the interaction by co-immunoprecipitation between these proteins in HeLa cells endogenously expressing Akt and BAG5. As shown in Figure 1H, Akt interacted with BAG5 as revealed by its presence in the lane in which immunoprecipitated BAG5 was electrophoresed, whereas it was barely detectable in the control lane in which a non-related antibody was used as control during the immunoprecipitation.

### 2.2. BAG5 Switches Akt Monoubiquitination to Polyubiquitination and Promotes degradation

Evidence that guided the potential involvement of BAG5 as a regulator of Akt stability emerged from initial experiments in which the interaction between Akt and BAG5 was characterized in mammalian cells. We found that different constructs of BAG5 reduced the expression of cotransfected Akt (see total cell lysates, TCL, in Figure 1F). Thus, considering that BAG proteins are recognized as chaperone regulators and BAG5, in particular, regulates protein stability via Hsp70 and modulates the activity of E3 ligases and autophagosome-related proteins [28,33,34,35], we wanted to evaluate the role of BAG5 as a potential modulator of Akt stability. In order to start exploring the mechanism by which BAG5 promoted Akt degradation, we assessed whether the kinase was targeted to ubiquitination via BAG5. In vivo ubiquitination assays using HeLa cells transfected with plasmids coding for HA-ubiquitin and Myc-BAG5 revealed that monoubiquitinated endogenous Akt was lost in response to BAG5 overexpression, which, in turn, promoted polyubiquitination of endogenous Akt (Figure 2A), as revealed by the Western blot with anti-HA antibodies that evidenced a smear of ubiquitin covalently bound to immunoprecipitated Akt (Figure 2A, middle panel). Intriguingly, even in the presence of MG132, a potent proteasome inhibitor, BAG5 caused Akt degradation (Figure 2B). Surprisingly, the levels of polyubiquitinated Akt in the presence of BAG5 were similar with or without added MG132 (Figure 2A). As a control, we revealed polyubiquitinated proteins in total cell lysates, which, as expected, increased by the effect of MG132 (Figure 2C), indicating the effectiveness of the proteasome inhibitor and further pointing to a proteasome-independent pathway by which BAG5 targets Akt to degradation, potentially involving Hsp chaperones.

We then assessed the effect of BAG5 on Akt expression in a situation known to put Akt stability at limit [36], consisting of starvation in conditions in which protein synthesis and Hsp90 chaperone activity were inhibited. These Akt-destabilizing conditions (ADC, serum-free media containing 5 μM cycloheximide and 1 μM 17-N-Allylamino-17-demethoxygeldanamycin) led to a time-dependent reduction in the expression of Akt, which was more noticeable when Myc-BAG5 was coexpressed (Figure 2D). Consistent with a destabilizing effect, BAG5 also reduced the levels of phospho-Akt (T450) without affecting the levels of endogenous Rac used as control (Figure 2D). In contrast, the BAG5 knockdown interfered with the effect of ADC on reducing Akt expression (Figure 2E), further confirming that BAG5 promotes Akt destabilization. Rac and Erk protein levels, used as controls, did not change in response to ADC regardless of changes in the expression of BAG5, either by overexpression or knockdown (Figure 2D,E). We then postulated that the effect of BAG5 on Akt was mediated by Hsp70, whose activity is known to be regulated by BAG5 [25,33], and which has a documented effect on Akt stability and function [24,37].

### 2.3. Hsp70 Mediates Akt Degradation Promoted by BAG5

BAG5 is a known modulator of Hsp70 activity [25,26], and this molecular chaperone has been reported as a controller of Akt stability [24]. Thus, we tested the hypothesis that BAG5 promotes Akt ubiquitination and degradation via the intervention of Hsp70 (Figure 3A). First, we evaluated the existence of a complex including Akt, BAG5, and Hsp70 in HEK293T by pull-down experiments of GST-BAG5. Consistent with previous results, HA-Akt interacted with BAG5 (Figure 3B, left panel). In addition, Hsp70 (tagged with EGFP) interacted with BAG5 and reduced the amount of Akt present in the pull-down (Figure 3B, left panel), suggesting that Hsp70 either competed with Akt in the interaction with BAG5 or contributed to its degradation. As expected, BAG5 caused a reduction in Akt levels, and this effect was more evident in the presence of Hsp70, as observed by Western blot using anti-HA antibodies in total cell lysates (Figure 3B, right panel). The expression of Hsp70 and BAG5 was confirmed in total cell lysates using anti-EGFP and GST antibodies. Consistent with previous studies [24], Hsp70 reduced the expression of Akt, both endogenous and transfected (detected with anti-HA antibodies), and this effect was more pronounced with increasing amounts of transfected Hsp70 (Figure 3C), which did not affect the expression of S6 used as control. Interestingly, the effect of Hsp70 as a promoter of Akt degradation was significantly reduced in BAG5 knockdown cells (Figure 3D). This effect was specific, as indicated by the equal expression of Erk and S6 in conditions in which Hsp70 was overexpressed both in control or BAG5 knockdown cells, suggesting that BAG5 contributes to the role of Hsp70 as a promoter of Akt degradation (Figure 3E).

### 2.4. Profile of Deubiquitinases and E3 Ligases Linked to BAG5 Expression in Breast, Lung, Uterine, and Ovarian Cancer Patients

Previous reports described the interaction between BAG5 and E3 ubiquitin ligases such as Parkin and CHIP [26,27]. On the other hand, CHIP cooperates with molecular chaperones such as Hsp70 to promote ubiquitination and degradation of selected protein targets [38,39,40,41]. Thus, we investigated the profile of deubiquitinases (DUBs) and E3 ligases coexpressed with BAG5 in cancer patients of the TCGA datasets in which the BAG5 gene was amplified (Figure 4A). To obtain an initial insight into the profile of DUBs and E3 ligases, we mined the TCGA cancer datasets, looking for those with a higher correlation with BAG5 expression. We identified the top ten most correlated DUBs and E3 ligases (Figure 4B,C). Eight out of ten DUBs and five out of ten E3 ligases were also essential in breast, uterine, lung, and ovarian cancer cell lines by knock-out studies, as revealed by data mining of the Cancer Dependency Map datasets (DepMap platform, https://depmap.org/portal/, accessed on 12 November 2023; Figure 4B,C, underlined genes). Next, we analyzed whether a change in Akt protein expression was evident in tumors with *BAG5* overexpression compared to all other tumors in breast, uterine, lung, and ovarian cancer types [15]. As evidence of the potential effect of BAG5 on Akt stability, we found that tumors overexpressing BAG5 contained higher Akt levels in uterine and ovarian tumors (Figure 4D). Since BAG5 amplification correlated with Akt1 protein expression, we identified and compared the top ten correlated DUBs and E3 ligases with BAG5 in general and in amplified BAG5 cancer patients. Several DUBs and E3 ligases showed higher correlation coefficients with BAG5 when patients were segregated by BAG5 amplification (Figure 4E,F). In addition, we investigated which DUBs and E3 ligases were found by both criteria, general correlation and segregated correlation; those were *USP7*, *OTUB2,* and *USP51* as DUBs, and *RLIM*, *HECTD1*, *XIAP,* and *UBR3* as E3 ligases (Figure 4E,F, genes highlighted in bold fonts).

### 2.5. Phosphorylation of BAG5 by Akt Correlates with a Reduction in the Interaction between Them

Since the ability of BAG5 to promote Akt degradation was particularly effective in conditions that put Akt stability to the limit (Figure 2D), we hypothesized that Akt stimulation modulates its interaction with BAG5 as a consequence of the phosphorylation of this interactor (Figure 5A). To start addressing this possibility, HeLa cells transfected with HA-Akt and Myc-BAG5 were stimulated with hepatocyte growth factor (HGF), then Akt was immunoprecipitated with HA antibodies, and the presence of BAG5 bound to it was assessed by Western blot using anti-Myc antibodies. Controls included non-stimulated cells in which vehicle was added instead of HGF and cells in which only BAG5 or Akt were transfected. As shown in Figure 5B, BAG5 more effectively interacted with HA-Akt in non-stimulated cells (−), whereas the amount of BAG5 that interacted with HA-Akt in HGF-stimulated cells was reduced (Figure 5B, top Western blot). As expected, HGF effectively induced Akt and Erk activation, as demonstrated by their phosphorylation visualized in total cell lysates (Figure 5B). These results suggest that activation of Akt disrupts its interaction with BAG5. Consistent with previous results, the effect of BAG5 on Akt stability was less effective on cells grown in complete media in contrast to cells subjected to overnight starvation, conditions that did not affect the stability of GSK-3β used as control (Figure 5C). A possible mechanism that explains why the interaction between Akt and BAG5 is disrupted upon Akt stimulation is that Akt phosphorylates BAG5, resulting in a reduction of its affinity for the kinase. To assess this possibility, the phosphorylation of BAG5 was tested in HEK293T cells transfected with myristoylated-Akt, which has been demonstrated to be a constitutively active kinase [42] or either wild-type or kinase-dead variants. Controls included cells in which Akt was not transfected. Cells were incubated with radioactive orthophosphate, and phosphorylation of immunoprecipitated Myc-BAG5 was detected by autoradiography. As shown in Figure 5D, BAG5 was phosphorylated under basal conditions (in the absence of transfected Akt), and this phosphorylation was increased in cells expressing constitutively active myristoylated Akt or wild-type Akt but not in cells overexpressing the catalytically inactive mutant version of this kinase. By MS analysis, the first BAG domain has been detected as the most prominent phosphorylated site in BAG5 (https://www.phosphosite.org; accessed on 14 November 2023).

Thus, to know whether this domain was phosphorylated by Akt, we cotransfected myr-Akt with this and the BAG5 linker region constructs (both of them with similar molecular weight), fused to GST (Figure 5E), and analyzed the pull-downs with antibodies against substrates phosphorylated by Akt (psus-Akt). We detected the first BAG domain as the most prominent phospho-substrate of myr-Akt (Figure 5F). These observations placed BAG5 as an Akt substrate and support the hypothesis that unphosphorylated BAG5 preferentially interacts with unstimulated Akt.

### 2.6. BAG5 Modulates Akt Phosphorylation and HGF-Dependent Cell Migration

The maximal activation of Akt has been considered to be achieved by phosphorylation at Ser473, which is located at the kinase regulatory carboxyl-terminal domain. This phosphorylation is critical to allow the intervention of this kinase in multiple processes, including cell migration, and is usually monitored to detect its activation status [2,43]. Thus, considering the reciprocal regulation occurring between Akt and BAG5, we hypothesized that besides being a promoter of Akt degradation under adverse nutritional conditions, BAG5 might also regulate Akt activation and cell migration in conditions in which the stability of the kinase is not particularly compromised. In order to test this hypothesis, we explored the influence of BAG5 on Akt activity, evidenced by its phosphorylation at Ser473 and in cell migration tested by chemotactic assays.

In order to assess the effect of BAG5 on Akt phosphorylation in cells stimulated with HGF, HeLa cells were transfected with an amount of HA-BAG5 that would not affect Akt stability. Then, cells were stimulated with increasing doses of HGF, and the phosphorylation of Akt and Erk, used as control, was detected by Western blot. Akt phosphorylation at Ser473 was stimulated by HGF in a dose-dependent manner, reaching a peak at 10 ng/mL. The reason why higher concentrations did not cause stronger Akt phosphorylation might be due to Met receptor desensitization linked to its phosphorylation at Ser985 by PKC, which decreases Met tyrosine kinase activity [44]. Akt phosphorylation was attenuated when HA-BAG5 was transfected in conditions in which the expression of total Akt remained comparable (Figure 6A). The specificity of this effect is evident when comparing the influence of BAG5 on Akt and Erk phosphorylation. While BAG5 attenuated the phosphorylation of Akt, it did not affect the phosphorylation of Erk assessed in the same total cell lysates (Figure 6A). We then assessed the effect of knocking-down or overexpressing BAG5 on the phosphorylation status of Akt at Ser473 in cells grown in complete media. Consistent with the regulatory role of BAG5 on Akt activation, the phosphorylation of Akt was reciprocally affected by the expression of BAG5, whereas BAG5 knockdown increased Akt phosphorylation (Figure 6B); this posttranslational modification was progressively inhibited by increasing amounts of Myc-BAG5 (Figure 6C).

To determine the influence of BAG5 on cell migration, a complex cellular event, which in cells responding to HGF is known to involve Akt [2,43], migration of HeLa cells transfected with Myc-tagged BAG5 was tested in chemotaxis assays in Boyden chambers. BAG5 significantly reduced HeLa cell migration in response to HGF in chemotaxis assays without significantly altering the response to fetal bovine serum used as a positive control, given its effect on multiple signaling routes (Figure 6D).

Altogether, our findings suggest a mechanism of reciprocal regulation between BAG5 and Akt depicted in the model presented in Figure 6E. Accordingly, Akt interacts with BAG5 under resting unstimulated cell conditions. When growth factor withdrawal gets critical, BAG5 targets Akt to ubiquitination and degradation with the participation of the molecular chaperone Hsp70. On the contrary, upon cell stimulation, Akt is activated, controlling multiple processes, including cell migration, and escapes from BAG5-dependent regulation by phosphorylating it and reducing their mutual interaction. This reciprocal regulation depends on the levels of expression of BAG5, which, besides contributing to target Akt for degradation, also modulates Akt activation at protein levels that do not affect Akt expression (Figure 6E).

## 3. Discussion

Clinical studies in cancer patients have confirmed the importance of targeting the PI3K/Akt signaling pathway, which is frequently exacerbated due to activating mutations in genes encoding key components of this signaling route [45]. Our findings demonstrate a novel mechanism of regulation of Akt stability. Accordingly, we postulate a model in which Akt is degraded as a consequence of its interaction with BAG5, which, under growth factor withdrawal, interacts with Akt and targets this kinase to ubiquitination with the participation of the molecular chaperone Hsp70. This process could help to fine-tune the apoptotic threshold when the lack of growth factors gets critical but may be turned off by the dissociation of Akt from BAG5 in response to growth factors, resulting from the phosphorylation of BAG5 by Akt (depicted in the model presented in Figure 6E).

Our results suggest that BAG5 promotes a decrease in Akt synthesis or contributes to its degradation. We found that BAG5, mainly through an internal region containing a pseudo-BAG domain, interacts with the serine/threonine kinase domain of Akt. Full-length BAG5, as well as the minimal interaction region extending from N87 to P181, promotes a decrease in Akt expression. Moreover, the short interaction region of BAG5 reduces the expression of the individually expressed Akt kinase domain. Since BAG5 is known to interact with the molecular chaperone Hsp70 and E3 ligases [19,25] and, as we show here, overexpression of BAG5 induces Akt ubiquitination and degradation, we hypothesized that BAG5 regulates Akt degradation through a pathway that includes the participation of Hsp70. This is consistent with the reported mechanism of protein complexes formed by Hsp70 and some E3 ligases, acting as molecular switches that determine the degradation of Hsp70 partners [46]. As an initial insight into the deubiquitination/ubiquitination system linked to BAG5 expression in cancer patients in which the BAG5 is amplified, we identified a group of DUBs and E3 ligases, which are essential in cancer cells, preferentially linked to BAG5 overexpression. Consistent with the role played by BAG5 in the regulation of protein ubiquitination and chaperone systems, it has been well documented that the E3 ligase parkin dissociates from Hsp70 upon the interaction of CHIP with Hsp70; thus, CHIP serves as an indirect controller of Parkin activity [46]. BAG5, according to our results, captures Akt under starvation conditions and presents it to Hsp70, which, similarly to the action of CHIP, could be sensitive to BAG5, resulting in the ubiquitination and degradation of Akt. Consistent with this model, Dickey and colleagues demonstrated that small molecules that allosterically inhibit the Hsp70 ATPase activity decreased Akt levels [24]. Particularly, an increase in Hsp70 levels combined with inhibition of its ATPase activity renders Akt unstable and results in an overall decrease in its expression [24]. Our results also show that Hsp70 overexpression reduces Akt levels and further supports the participation of BAG5 in the system, as indicated by the reduced effect of Hsp70 in BAG5 knockdown cells, suggesting that BAG5, in fact, modulates the action of Hsp70 towards Akt. Since BAG5 promotes Hsp70 refolding activity [25], our observations extend the possibilities of how this chaperone is controlled by including BAG5 as a dual element able to modulate Hsp70 (as reported) but also bringing Akt to the complex. Intriguingly, different variants of Hsp70 have opposite effects on Akt stability, further emphasizing the existence of a refined mechanism of control of Akt stability [24].

The effect of BAG5 on Akt stability is regulated by growth factor signaling. We found that Akt interacts with BAG5 under stressful serum starvation conditions but dissociates in response to hepatocyte growth factor. This correlates with the phosphorylation of BAG5, suggesting a mechanism of action linked to the signaling status of the cell. Our results contribute to reconciling the apparently conflicting observations on the role of BAG5 on cell viability, as it has been demonstrated that it either promotes cell death or survival [26,47].

Several differences can be highlighted among reported mechanisms of control of Akt ubiquitination/degradation and the mechanism postulated in this study. Here, we described a mechanism dynamically regulated either by critical serum starvation or growth factor stimulation. In contrast, fully active Akt, phosphorylated at Ser473, is preferentially ubiquitinated and targeted to degradation by the E3 ligase Mulan [48]. In addition, it is known that Mulan is a mitochondrial protein with its ring finger E3-ligase active domain exposed to the cytosol [49], implying that Mulan-dependent ubiquitination of Akt occurs associated with the cytosolic face of the mitochondria.

As in the case of BAG5, TRB3, another endogenous inhibitor of Akt, interacts with this kinase under poor nutritional conditions. As mentioned, the interaction between Akt and BAG5 occurs under starvation conditions; similarly, TRB3, an inhibitor that interferes with the phosphorylation of Akt at Thr308, preferentially interacts with Akt in the liver of fasted mice, as detected in immunoprecipitates from whole liver [50]. Although TRB3 inhibits Akt without affecting its expression, it regulates other proteins acting as an adaptor, leading to its degradation. In particular, it interacts with acetyl-coenzyme A carboxylase and regulates its degradation via association with the E3 ligase COP1 [51]. According to our results, BAG5 has a dual influence on Akt; it targets the kinase to degradation under critical serum starvation but also affects its activation in conditions in which Akt stability is not altered. This duality might be important in the regulation of Akt in cancer patients overexpressing BAG5, which we found to be statistically linked to higher Akt expression. This regulation could be related to the decrease in Akt monoubiquitination caused by BAG5, as it has been reported that this posttranslational modification contributes to Akt activation by helping to recruit the kinase to the plasma membrane during the activation process [8]. In addition, Akt reactivation, as a mechanism of drug resistance in preclinical models of breast cancer, has been linked to its ubiquitination [13]. The mechanism of Akt regulation by BAG5 is reciprocal, as evidenced by the phosphorylation of BAG5 by Akt, which correlates with a reduction of the interaction between these proteins. Thus, this reciprocal mechanism fine-tunes the regulation of Akt under critical and stimulated conditions, as depicted in the model shown in Figure 6E.

Altered mechanisms related to Akt regulation are involved in cancer and metabolic diseases. Here, we show that BAG5 overexpression correlates with increased Akt expression at the protein level in tumors of uterine and ovarian cancer patients. Considering the multiple tumor-derived factors present in the tumor microenvironment, the effect of BAG5 in these settings might be linked to its effect on Akt activity and might be regulated by the effects of coexpressed deubiquitinases and E3 ligases, which warrant future investigations in preclinical models. Given that regulation of Akt expression is of clinical interest in oncology, the identification of coexpressed proteins linked to ubiquitin-linked pathways raises hypothesis-driven questions that set the basis for future investigations. Finding the mechanisms that determine the threshold of Akt stability will contribute to optimizing anti-oncogenic therapies, many of which are particularly oriented to target the PI3K/Akt pathway in cancer cells. Our results showing that BAG5 levels correlate with Akt stability opens the possibility for future studies in which modulation of BAG5 expression and activity would increase the effectiveness of pharmacological strategies targeting the PI3K/Akt signaling pathway.

## 4. Materials and Methods

### 4.1. Yeast Two-Hybrid Screening

The mouse Akt1-K179M kinase dead domain subcloned into pGB3, in frame with the DNA–binding domain of GAL4, was used as bait to screen a human fetal brain cDNA library (Clontech) by the yeast two-hybrid system, using the Matchmaker System III (Clontech) following the manufacturer’s instructions with some modifications, as previously described [52]. Putative interacting clones were selected in media lacking His/Leu/Trp and by their positive reaction for alpha-galactosidase using X-alpha Gal assay as a visible indicator of activity. The specificity of the interaction was confirmed using p53 and SV40 large T antigens as bait and prey, respectively.

### 4.2. Plasmids and DNA Constructs

The mouse Akt1-K179M kinase dead domain >KDD-AKT(150-408)

FEYLKLLGKGTFGKVILVKEKATGRYYAMKILKKEVIVAKDEVAHTLTENRVLQNSRHPFLTALKYSFQTHDRLCFVMEYANGGELFFHLSRERVFSEDRARFYGAEIVSALDYLHSEKNVVYRDLKLENLMLDKDGHIKITDFGLCKEGIKDGATMKTFCGTPEYLAPEVLEDNDYGRAVDWWGLGVVMYEMMCGRLPFYNQDHEKLFELILMEEIAFPRTLGPEAKSLLSGLLKKDPTQRLGGGSEDAKEIMQHRFF was subcloned by PCR using the following primers: >Akt kdom 5’ BamHI ataGGATCCtttgagtacctgaaactact, >Akt kdom 3’ EcoRI ataGAATTCaaagaaccggtgctgcatga. BAG5 cDNA was subcloned from the yeast two-hybrid prey pACT2 vector into pCMV-HA, pCMV-Myc, and pCEFL-EGFP mammalian expression vectors. For pull-down experiments, full-length BAG5 or constructs including the linker region alone (BAG5 N87-P181) or extended to the amino terminus (BAG5 M1-P181) or the carboxyl terminus (BAG5 N87-Y447), containing the first BAG domain or the last three BAG domains, respectively, were subcloned by PCR, in frame with the sequence coding for GST into the mammalian expression vector pCEFL-GST. The following primers were used: BAG5-5’BamHI ataGGATCCatggatatgggaaaccaacatcc, BAG5-5’N87-BamHI ataGGATCCaaccacccacaccggattg, BAG5-3’P181-EcoRI ataGAATTCtcaaggatgtgcatcctcggaaagcg, BAG5-3’EcoRI ataAATTCtcagtactcccattcatcag. The shRNA for BAG5 was generated based on the sequence of the hairpin >shRNAhBAG5-669-HP_63610 TGCTGTTGACAGTGAGCGCAGGTATCACACTTTAACCAAATAGTGAAGCCACAGATGTATTTGGTTAAAGTGTGATACCTTTGCCTACTGCCTCGGA and cloned into pENTR vector. Plasmid coding for EGFP-Hsp70 [53], from Lois Greene, Ph.D. NHLBI, NIH, Bethesda, MD USA, was obtained from Addgene (plasmid 15215). All other plasmids have been described previously [54,55].

### 4.3. Cell Culture and Transfection

The Human Embryonic Kidney 293T (HEK293T) and HeLa cells were maintained in Dulbecco’s modified Eagle’s medium (DMEM; Invitrogen, catalog 15240112) supplemented with 10% fetal bovine serum (FBS), 100 μg/mL penicillin and 100 μg/mL streptomycin (Gibco) at 37 °C in a humidified atmosphere with 5% CO_2_. Cells were transfected transiently for 48 or 72 h with Polyfect (Qiagen, 301105) according to the instructions of the manufacturer.

### 4.4. Immunoprecipitation, Pull-Down, and Western Blot Assays

Whole-cell lysates were obtained from cells grown in 10 cm Petri dishes, washed with PBS (pH 7.4), and lysed with ice-cold TNTE lysis buffer (50 mM Tris [pH 7.5], 150 mM NaCl, 1% Triton X-100, and 5 mM EDTA) containing protease inhibitors (1 mM phenylmethylsulfonyl fluoride, 10 μg/mL leupeptin, and 10 μg/mL aprotinin) and phosphatase inhibitors (20 mM β-glycerophosphate, 1 mM sodium vanadate, and 1 mM sodium fluoride). For immunoprecipitations, whole-cell lysates were centrifuged at 14,000 rpm in a refrigerated bench-top microcentrifuge (Eppendorf, Hamburg, Germany) for 10 min at 4 °C. Supernatants were incubated with specific primary antibodies overnight at 4 °C. Immunoprecipitates were harvested by incubation with protein A/G agarose beads for 2 h at 4 °C in a rocking platform and washed five times with lysis buffer. Then, boiled in 1X Laemmli sample buffer for 5 min. Immunoprecipitates and 20 μg of total proteins were separated by SDS-PAGE (using 6% or 10% resolving gels) and transferred onto polyvinylidene fluoride membrane (PVDF, Millipore IPV00010) for 2.5 h at 320 mAmp. Protein concentrations in total cell lysates were determined using a DC protein assay kit (Bio-Rad, Hercules, CA, USA). Membranes were blocked with 5% non-fat dry milk in TBS-T (0.05% Tween-20 in Tris-buffered saline, pH 7.4) for 1 h at room temperature, then washed twice with TBS-T and incubated with appropriate antibodies. For most experiments, primary antibodies were used at 1:5000 dilution in TBS-T incubated overnight at 4 °C. Membranes were rinsed with TBS-T and incubated with specific secondary antibodies conjugated to HRP (1:10,000–1:30,000) in a blocking solution for 1 h at room temperature and revealed using Immobilon Western chemiluminescent substrate (Millipore, Burlington, VT, USA, WBKLS0500). The interaction between transfected BAG5 and Akt, either full length or the indicated domains, was assessed in HEK293T cells using expression constructs of Myc- or GST-tagged BAG5 and HA-tagged Akt1 or its kinase-dead domain (Akt-KDD, K179M) fused to EGFP. For comparison, in some experiments, HA-tagged Erk was cotransfected with HA-Akt and GST-BAG5 constructs adjusted to reach expression levels similar to HA-Akt. Two days after transfection, GST-BAG5 was isolated by pull-down using 40 μL of glutathione sepharose 4B beads for 2 h at 4 °C, with rotation, and interacting Akt was revealed by Western blot using anti-HA or GFP antibodies. For experiments in which Akt was immunoprecipitated with HA antibodies, interacting Myc-BAG5 was revealed by Western blot using anti-Myc antibodies. In order to test for interactions of BAG5 with Hsp70, cells were cotransfected with GST-BAG5, HA-Akt, and EGFP-Hsp70; control experiments included GST instead of GST-BAG5. After 36 h, cells were incubated for 12 h in serum-free media, then 20 μM MG132 was added, and incubation continued for an additional 4 h. Cells were lysed in TNTE buffer supplemented with protease and phosphatase inhibitors, and pull-down was performed as indicated above. The potential interaction between endogenous BAG5 and Akt was assessed by immunoprecipitation of BAG5 followed by Western blot against Akt with lysates obtained from serum-starved Hela cells treated with MG132 for 12 h. To evaluate the effect of Akt stimulation on its interaction with BAG5, cells were serum-starved overnight in DMEM and either stimulated with 10 ng/mL HGF for 15 min or the corresponding vehicle. Primary antibodies with the following specificity were used: Akt1 (Sigma, St. Louis, MO, USA, P2482), phospho-Akt Thr450 (Cell signaling, catalog 12178), phospho-Akt (Ser473, catalog sc-7985), phospho-Erk1/2 T202/Y204 (Cell Signaling, catalog 9191), S6 ribosomal protein (Cell Signaling, catalog 2217), GSK3β (Cell Signaling, catalog 9323), BAG5 (Imgenex, Odisha, India, IMG-5678), Rac1 (BD-Biosciences, Franklin Lakes, NJ, USA, catalog 610651), HA (Covance, MMS-101R) and GFP (Santa Cruz, sc-9996), Myc (Sigma M4439), Erk2 (Santa Cruz, CA, USA, sc-154) and GST (Santa Cruz, sc-154). Protein A- (catalog 16-125) and protein G-agarose (catalog 16-266) were purchased from Millipore. The glutathione-Sepharose 4B was obtained from GE Healthcare Life Sciences, Marlborough, MA, USA (catalog 17-0756-05).

### 4.5. Akt Stability and In Vivo Ubiquitination Assay

To explore the effect of BAG5 and Hsp70 on Akt stability, cells were transfected with increasing amounts of plasmid coding for these proteins and HA-Akt, as indicated in figure legends. In some experiments, a knockdown approach was used to assess the contribution of BAG5 to Akt degradation. In these cases, a specific BAG5 shRNA in pENTR was used, whereas controls were transfected with pENTR containing a non-related, unspecific sequence. The effect of Akt activation on its sensitivity to be regulated by overexpression of BAG5 was assessed by comparing the effect of increasing concentrations of BAG5 in serum-starved cells or cells stimulated with 10% serum. In some experiments, cells were incubated under what we called Akt-destabilizing conditions consisting of serum-free media supplemented with 5 μM CHX to inhibit protein synthesis and 1 μM 17AAG for different times up to 24 h. To inhibit proteasome activity, cells were incubated with 20 μM MG132 for 4 h. For in vivo ubiquitination assays, HeLa cells were transfected with plasmids coding for HA-Ubiquitin and Myc-BAG5. After 36 h, cells were starved with serum-free media for 12 h. Then, it was incubated with 20 μM MG132 for 4 h and lysed in TNTE buffer. Endogenous Akt was immunoprecipitated with a specific antibody, as described above. Samples were boiled 5 min to elute proteins from beads in 1X Laemmli sample buffer. Immunoprecipitates and total cell lysates were resolved by SDS-PAGE using 6% gels and transferred onto the PVDF membrane for immunoblotting analyses. Ubiquitinated Akt was visualized using anti-HA antibodies to reveal HA-tagged ubiquitin covalently bound to immunoprecipitated Akt. Hsp90 inhibitor 17-N-Allylamino-17-demethoxygeldanamycin (17AAG) and cycloheximide were purchased from Sigma, and MG132 was from Calbiochem.

### 4.6. TCGA Data Mining Guided by BAG5

To select TCGA cancer studies with overexpressed BAG5, we analyzed *BAG5* mRNA expression (RSEM (Batch normalized from Ilumina HiSeq_RNASeqV2)) from cBioPortal (https://www.cbioportal.org/; accessed on 25 October 2023) and selected those with a correlation between amplification and high expression. Based on the populations with and without amplified BAG5, we analyzed Akt1 protein expression (RPPA). Guided by BAG5 high expression, coexpression coefficients were searched for deubiquitinases and E3 ligases.

Deubiquitinases and E3 ligases were analyzed in the DepMap platform (https://depmap.org/portal/; accessed on 12 November 2023) to identify if they were essential in breast, uterine, lung, and ovarian cancer cell lines when they were knock-out by CRISPR (DepMap_Public_23Q2+Score,_Chronos).

### 4.7. BAG5 Phosphorylation Assays

To test whether Akt played a role in the phosphorylation of BAG5, HEK293T cells in 10 cm dishes were transfected with Myc-BAG5 and control empty plasmid or plasmids coding for HA-Akt, Myr-Akt, or Akt-K179M, corresponding to wild type, constitutively active and dominant negative versions of Akt. Two days after transfection, serum-starved cells were metabolically labeled for 4 h at 37 °C in 5 mL of Pi-free DMEM containing 0.1% (*w*/*v*) BSA and 100 μCi/mL ^32^Pi. After labeling, cells were washed with ice-cold PBS and lysed in 50 mM Tris, pH 8.0, 100 mM NaCl, 1% (*v*/*v*) NP40, 1% (*w*/*v*) Na deoxycholate, 0.1% (*w*/*v*) SDS, containing 20 mM NaF, 10 mM Na pyrophosphate, 5 mM EDTA, 10 μg/mL aprotinin, 10 μg/mL leupeptin, 10 μg/mL soybean trypsin inhibitor, 10 μg/mL pepstatin, 10 μg/mL benzamidine, 1 mM PMSF and 1 μM okadaic acid. Then, cell lysates were pre-cleared by being incubated with 2% (*v*/*v*) protein G plus/protein A-agarose for 1 h at 4 °C and immunoprecipitation of Myc-tagged BAG5 was performed by adding a high-affinity Myc antibody (1 μL) and 2% (*v*/*v*) protein G plus/protein A-agarose followed by incubation overnight at 4 °C. After washing of the sepharose-bound immune complexes in solubilization buffer lacking protease inhibitors, ^32^P-labeled phospho-BAG5 were eluted in Laemmli sample buffer for 1 h at 48 °C and resolved by SDS-PAGE (10% resolving gel) after loading equal amounts of protein in each lane prior to visualization in a Typhoon TRIO Variable Mode Imager (GE Healthcare).

### 4.8. Akt Phosphorylation

To explore the effect of BAG5 knockdown or overexpression on Akt activation, in conditions in which BAG5 does not affect Akt stability, cells were transfected with increasing amounts of BAG5-specific shRNA or Myc-BAG5 and maintained in 10% FBS-supplemented media. Then, phosphorylation of Akt at Ser473 was detected by Western blot in cells starved for 12 h and, in some experiments, treated with increasing concentrations of HGF, as indicated in the corresponding figures.

### 4.9. Chemotaxis Assays

The effect of BAG5 on cell migration was assessed by chemotaxis assays. Confluent cultures of HeLa cells, transfected with control plasmid or Myc-tagged BAG5, were used for chemotaxis assays performed in Boyden chambers essentially as previously described [56,57]. Briefly, bottom wells were filled with serum-free DMEM containing 0.1% bovine serum albumin (used as a vehicle) and either 10 μg/mL HGF or 10% FBS and covered with a polycarbonate filter (8 μm pore; Neuro Probe, Gaithersburg, MD, USA), previously coated with 2% gelatin (from bovine skin, type B, Sigma). The chamber was assembled following instructions provided by the manufacturer. Then, serum-starved transfected HeLa cells, suspended in serum-free media supplemented with 0.1% bovine serum albumin, were placed in the upper wells (50,000 cells/well) of a 48-well chamber. Cells were left to migrate for 6 h. Then, the filters were fixed with methanol and stained with crystal violet, and nonmigrating cells remaining at the upper side of the filter were removed with a cotton swab. Filters were placed on a glass slide and scanned. Densitometric quantification of migrating cells was determined with the ImageJ software (https://imagej.net/ij/, accessed on 1 September 2023).

### 4.10. Statistical Analysis

All experiments were repeated at least three independent times. Results were analyzed by one-way ANOVA and student *t*-test using the GraphPad software, version 5, and represented as means +/− standard errors, indicating statistically significant differences when <0.05.

## Figures and Tables

**Figure 1 ijms-24-17531-f001:**
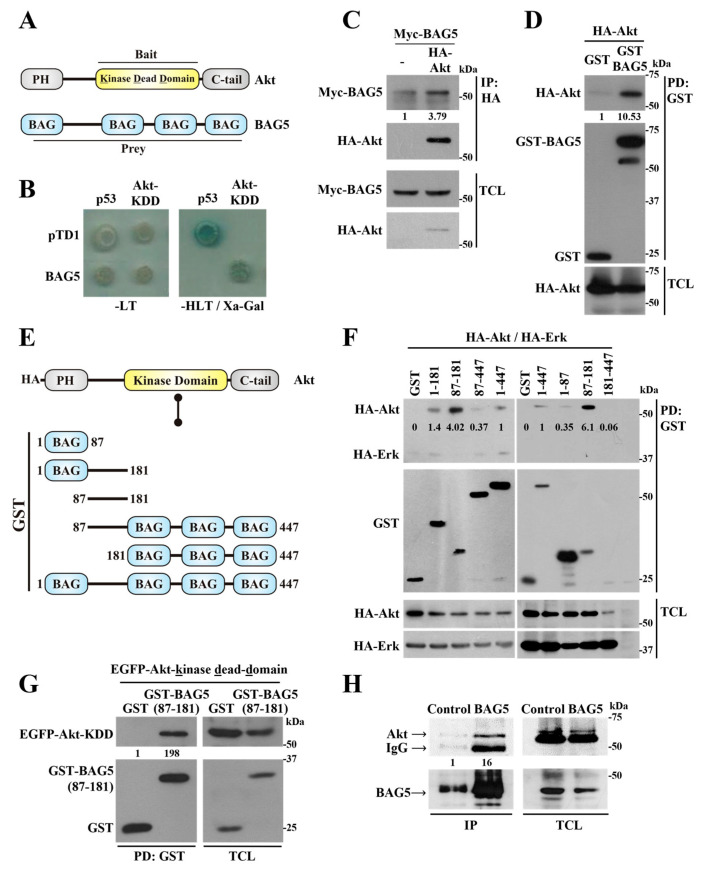
BAG5 interacts with Akt. (**A**) Schematic representation of Akt and BAG5 structures indicating Bait and Prey and their corresponding interacting domains. (**B**) BAG5 and Akt interact in the yeast two-hybrid system. Akt-KDD (dominant negative mutant of the catalytic domain of Akt) was used as bait to screen a human fetal brain cDNA library in a yeast two-hybrid system. A full-length clone of BAG5 was identified as a novel Akt-interactor protein. The specificity of the interaction between BAG5 and Akt was determined using p53 and pTD1 (T antigen) as negative control bait and prey, respectively, whereas the known interaction between p53 and T antigen served as positive control of the system. All yeast grew on media lacking leucine and tryptophan (-LT, which selects for the presence of the plasmids, left), but only those in which interactions occurred grew in media lacking histidine, leucine, and tryptophan and were positive for the activity of α-galactosidase (-HLT + Xα-Gal, right). (**C**,**D**) Transfected Akt and BAG5 interact in mammalian cells. HEK293T cells were cotransfected with HA-Akt and Myc-BAG5 (**C**) or with HA-Akt and GST-BAG5 (**D**), as indicated. Total cell lysates (TCL) were either immunoprecipitated using anti-HA antibodies ((**C**), IP:HA) or pulled down using glutathione beads ((**D**), PD:GST). BAG5 that coimmunoprecipitated with Akt was detected with Myc-specific antibodies (**C**), whereas Akt that interacted with pulled-down BAG5 was detected with HA-specific antibodies (**D**); the expression of both proteins was confirmed in total cell lysates ((**C**,**D**), TCL). (**E**) Schematic representation of BAG5 constructs used to map the interaction with Akt. (**F**) BAG5 linker region (N87-P181) interacts with Akt but not with Erk. The illustrated BAG5 constructs, including the N87-P181 linker region alone or extended towards the first or last three BAG domains fused to GST, were used to identify the region in BAG5 that interacts with Akt, the specificity of the interaction was confirmed using comparison against Erk. GST was used as a negative control. GST-BAG5 pull-down assays were performed in HEK293T cells transiently cotransfected with HA-Akt and HA-Erk and either one of the indicated BAG5 constructs. The ability of either Akt or Erk (both tagged with HA) to interact with BAG5 constructs was revealed by Western blot using anti-HA antibodies. (**G**) BAG5-linker region (N87-P181) interacts with the kinase-dead domain of Akt. The interaction between GST-BAG5-linker region (N87-P181) and EGFP-Akt-KDD was determined using pull-down assays using glutathione resin, the fraction of Akt-KDD that interacted with BAG5-linker region was revealed by Western blot using EGFP antibodies (left panel), whereas its expression was confirmed in total cell lysates (TCL, right panel). (**H**) Endogenous Akt interacts with BAG5. HeLa cell lysates were used to immunoprecipitate endogenous BAG5, and the presence of Akt bound to it was revealed by Western blot in the immunoprecipitate (IP), whereas its expression was confirmed in TCL. As a negative control, (**C**) a non-related antibody (anti-PAK) was used instead of the anti-BAG5 antibody. The arrows indicate the bands corresponding to Akt, BAG5, or IgG heavy chain. The numbers below the top panels (**C**,**D**,**F**,**G**,**H**) indicate the normalized densitometric value with respect to the condition indicated at 1.

**Figure 2 ijms-24-17531-f002:**
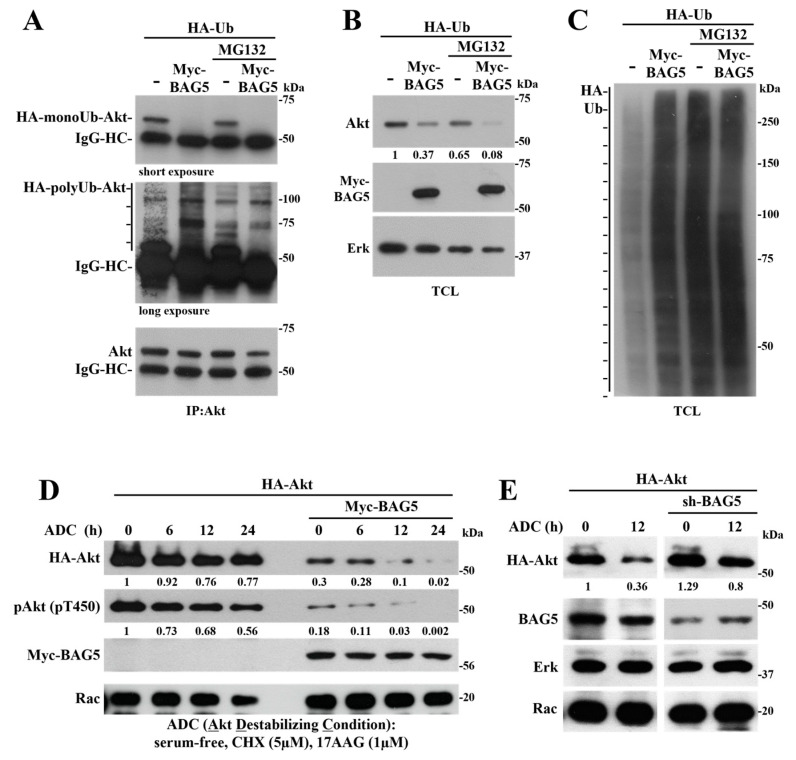
BAG5 promotes Akt ubiquitination and degradation. (**A**) BAG5 controls Akt ubiquitination. HeLa cells were transfected with HA-Ubiquitin and Myc-BAG5. Thirty-six hours after transfection, cells were serum-starved overnight and then left untreated or treated with 20 μM of MG132 for 6 h. Akt was immunoprecipitated from total cell lysates, and the presence of ubiquitinated Akt was detected as a high molecular smear in the immunoblots using anti-HA antibodies. (**B**) Akt protein levels are regulated by changes in the expression of BAG5. Increasing amounts of Myc-BAG5 induce a decrease in Akt protein expression, regardless of the effect of MG132, without altering the expression of Erk detected in the same total cell lysates. (**C**) Protein ubiquitination in total cell lysates of HeLa cells transfected with BAG5 and treated with MG132, as indicated in (**A**). (**D**) BAG5 accelerates Akt degradation under Akt destabilizing conditions (ADC). HeLa cells transiently transfected with HA-Akt and Myc-BAG5 were treated with 5 μM of cycloheximide (CHX) and 1 μM of 17-N-Allylamino-17-demethoxygeldanamycin (17AAG) in serum-free media, a condition known to destabilize Akt, during the indicated times. The expression of Akt, phospho-Akt (T-450), and BAG5 was detected by Western blot in total cell lysates (anti-HA for Akt and anti-Myc for BAG5), as well as the expression of Rac, used as loading control. (**E**) BAG5 knockdown attenuates the degradation of Akt under ADC treatment. HeLa cells were transfected with HA-Akt and shRNA-BAG5 for 60 h, followed by ADC treatment for 12 h, as indicated. Akt, BAG5, Erk, and Rac protein levels were analyzed by Western blot with the indicated antibodies. Rac and Erk served as a loading control. The numbers below the top panels (**B**,**D**,**E**) indicate the normalized densitometric value with respect to the condition indicated at 1.

**Figure 3 ijms-24-17531-f003:**
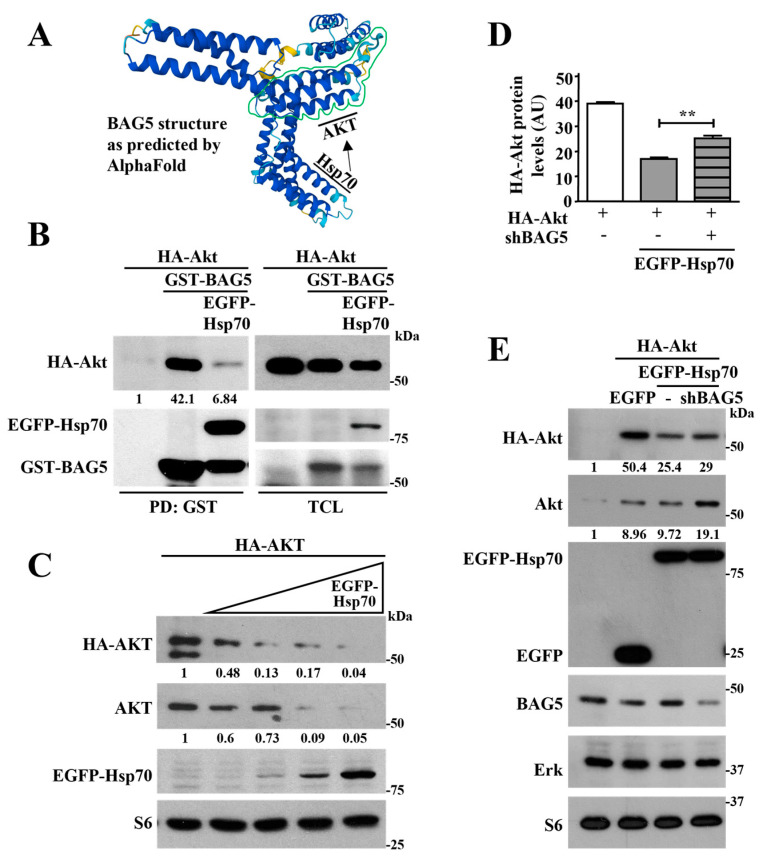
Hsp70 mediates Akt degradation promoted by BAG5. (**A**) Hypothetical model showing that BAG5 contributes to Akt regulation by Hsp70. BAG5 structure is shown as predicted by AlphaFold. (**B**) BAG5 interacts with Akt and Hsp70. HEK293T cells were transiently transfected with HA-Akt either in the presence or absence of GST-BAG5 alone or with EGFP-Hsp70, as indicated. GST-BAG5 was isolated by pull-down (PD:GST) from total cell lysates (TCL) using glutathione beds. The interaction of HA-Akt and EGFP-Hsp70 proteins with BAG5 was detected in the pull-down, whereas their expression was confirmed in total cell lysates (TCL), using anti-HA or anti-EGFP antibodies, respectively. (**C**) Hsp70 promotes Akt degradation. HeLa cells transiently transfected with HA-Akt, and increasing amounts of EGFP-Hsp70 were analyzed by Western blot of total cell lysates for the expression of Akt (anti-HA and anti-Akt), Hsp70 (anti-EGFP), and S6, 48 h post-transfection. Anti-S6 immunoblot served as a loading control. (**D**) Graph illustrates the densitometric analysis of HA-Akt expression in the presence or absence of Hsp70 and the effect of BAG5 knockdown on the effect of Hsp70. Bars represent the mean value ± SEM of three independent experiments, **, *p* < 0.01. (**E**) Hsp70 involves BAG5 to promote Akt degradation. HeLa cells were transiently transfected with shRNA-BAG5 for 24 h; then, cells were transfected again with HA-Akt and EGFP-Hsp70 for 48 h more to achieve 72 h of shRNA BAG5 effect. Akt, Hsp70, BAG5, Erk, and S6 levels were analyzed by immunoblot using antibodies against HA or EGFP for transfected Akt and Hsp70, respectively, or antibodies that recognize endogenous BAG5, Akt, Erk, or S6 as indicated. Erk and S6 immunoblots served as loading controls. The numbers below the top panels (**B**,**C**,**E**) indicate the normalized densitometric value with respect to the condition indicated at 1.

**Figure 4 ijms-24-17531-f004:**
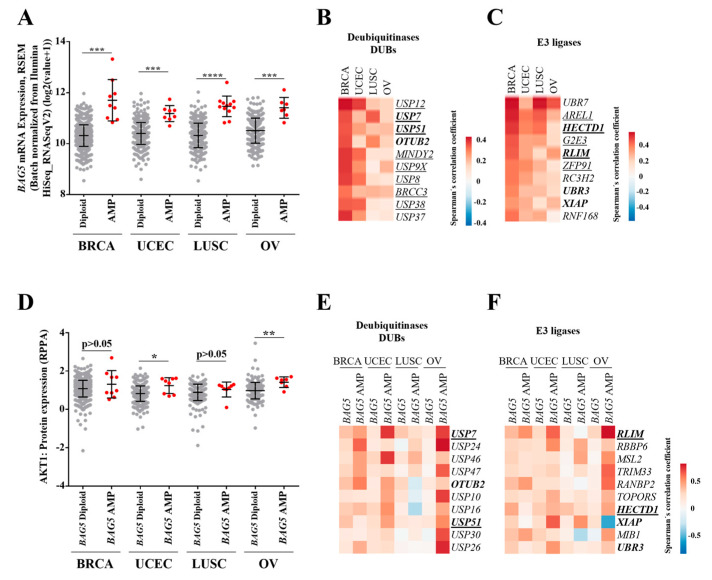
Profile of Deubiquitinases and E3 ligases linked to BAG5 expression in BRCA, UCEC, LUSC, and OV TCGA cancer patients. (**A**) BAG5 mRNA expression in BRCA, UCEC, LUSC, and OV TCGA studies in diploid and BAG5 amplified samples. Mean ± S.E.M values. ***, *p <* 0.001, ****, *p* < 0.0001. Unpaired *t* test with Welch’s correction. (**B**) Heatmap for the top ten DUBs correlated genes with BAG5 in BRCA, UCEC, LUSC, and OV cancer studies. (**C**) Heatmap for the top ten E3 ligases correlated genes with BAG5 in BRCA, UCEC, LUSC, and OV cancer studies. (**D**) Akt1 protein expression according to diploid and BAG5 amplified samples. Mean ± S.E.M values. *, *p* = 0.02, **, *p* = 0.0048. Unpaired *t* test with Welch’s correction. (**E**) Highest correlated DUBs with BAG5 in BAG5 amplified samples compared to general correlation. (**F**) Highest correlated E3 ligases with BAG5 in BAG5 amplified samples. Spearman’s correlation coefficients are calculated in cBioPortal. Genes in bold represent DUBs and E3 ligases correlated with BAG5 found in all patients or in BAG5 amplified patients. Underlined genes represent essential DUBs and E3 ligases in breast, uterine, lung, and ovarian cancer cell lines from the CRISPR effect in the DepMap platform.

**Figure 5 ijms-24-17531-f005:**
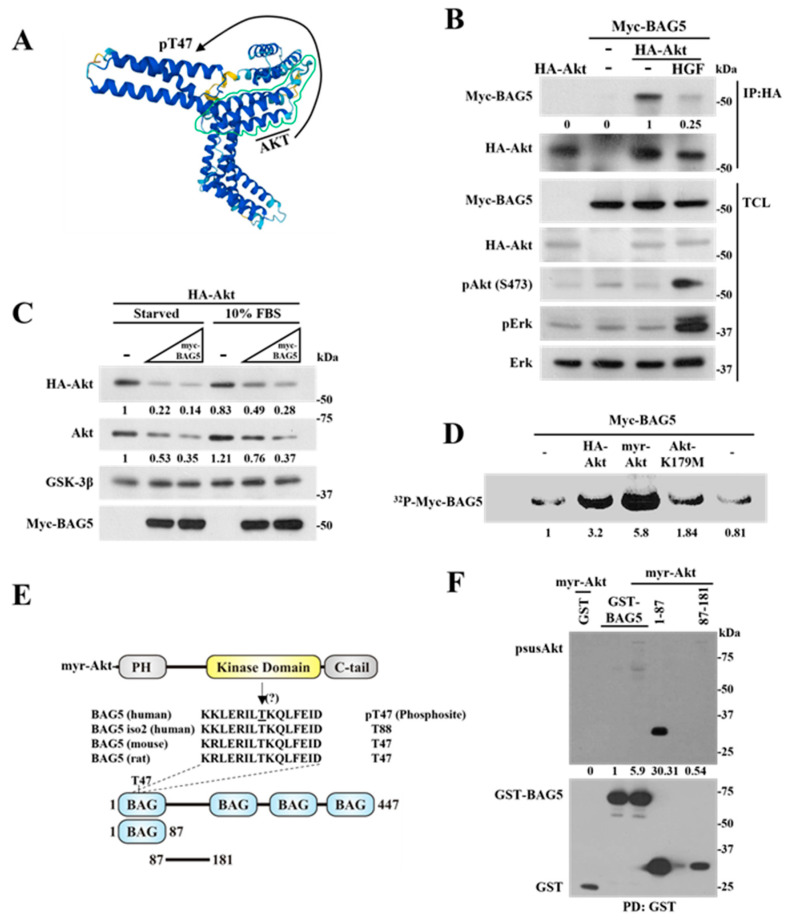
Phosphorylation of BAG5 by Akt correlates with a reduction in the interaction between them. (**A**) Hypothetical model showing the phosphorylation of BAG5 by Akt. The structure of BAG5 is shown as predicted by AlphaFold, and the indicated phosphorylation site is the most prominent in the phosphoproteomic analysis available at https://www.phosphosite.org/ (accessed on 14 November 2023). (**B**) Akt activation attenuates its interaction with BAG5. HeLa cells transfected with HA-Akt, Myc-BAG5, or both were incubated overnight in serum-free media and then stimulated, or not, with HGF for 15 min, as indicated. Akt was immunoprecipitated from total cell lysates using an anti-HA antibody (IP:HA), and interacting-BAG5 was detected by anti-Myc Western blot. The expression of Akt and BAG5 was confirmed in total cell lysates (TCL) using anti-HA and anti-Myc antibodies, respectively. The effect of HGF on Akt and Erk activation was detected with phospho-specific antibodies. Expression of Erk in total cell lysates served as a loading control. (**C**) Akt degradation induced by BAG5 preferentially occurs under serum starvation conditions. HeLa cells were cotransfected with HA-Akt and increasing amounts of Myc-BAG5. Thirty-six hours post-transfection, cells were incubated with serum-free or serum-supplemented media for an additional 12 h. Expression of Akt, GSK-3β, and BAG5 was analyzed by immunoblot in total cell lysates. GSK-3β served as a loading control. (**D**) Akt phosphorylates BAG5. Cells were transfected with Myc-BAG5 and one of the following Akt constructs: HA-Akt (wild type), myr-Akt (N-myristoylated-Akt, constitutively active), or kinase-negative mutant Akt (Akt-K179M). Twenty-four hours post-transfection, cells that were starved of serum overnight were metabolically labeled with ^32^P orthophosphate. BAG5 was isolated by immunoprecipitation with anti-Myc antibodies, and its phosphorylation was detected by autoradiography. (**E**) Schematic representation of BAG5 phosphorylation site at the first BAG domain, which is phylogenetically conserved (https://www.phosphosite.org/, accessed on 14 November 2023). The constructs represented at the bottom were used to investigate their potential phosphorylation by Akt. (**F**) Effect of myr-Akt on the phosphorylation of BAG5 and the indicated constructs, corresponding to the first BAG domain and the linker region (as shown in (**E**)). Phospho-Akt-substrates were detected with anti-psusAkt antibodies, and isolated constructs were revealed with anti-GST antibodies. The numbers below the top panels (**B**–**D**,**F**) indicate the normalized densitometric value with respect to the condition indicated at 1.

**Figure 6 ijms-24-17531-f006:**
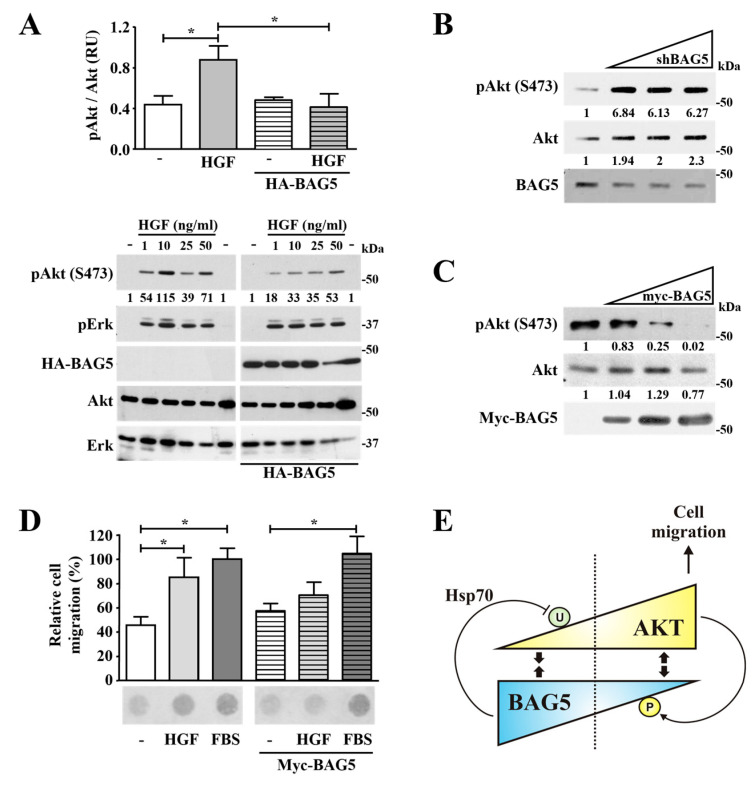
BAG5 modulates Akt activation and HGF-dependent cell migration. (**A**) BAG5 attenuates the activation of Akt in response to HGF without affecting the activation of Erk. HeLa cells transfected with the control vector, or HA-BAG5, were serum starved and stimulated with increasing amounts of HGF for 15 min. Then, activation of Akt and Erk was detected by Western blot using phospho-specific antibodies, whereas their expression was confirmed using antibodies that detect endogenous proteins. The expression of transfected BAG5 was confirmed using anti-HA antibodies. Graph at the top represents the analysis of Akt phosphorylation determined in three independent experiments. Mean ± S.E.M values. *, *p* < 0.05. (**B**,**C**) BAG5 expression reciprocally correlates with Akt activation in conditions in which Akt stability is not affected. (**B**) HeLa cells transiently transfected with increasing amounts of shRNA-BAG5 (**B**) or Myc-BAG5 (**C**) were used to detect Akt activation by Western blot in total cell lysates with a phospho-Akt (Ser473) specific antibody. The expression of total Akt, as well as transfected Myc-BAG5, was confirmed in the same samples, as indicated. The numbers below the top panels (**A**–**C**) indicate the normalized densitometric value with respect to the condition indicated at 1. (**D**) BAG5 diminishes HGF-dependent chemotactic cell migration. HeLa cells transfected with BAG5 or control plasmid were subjected to chemotaxis assays in Boyden chambers in which they were stimulated with HGF or 10% FBS, as indicated. Graph represents the analysis of relative cell migration determined in three independent experiments. Mean ± S.E.M values. *, *p* < 0.05. A representative result is shown at the bottom of the graph. (**E**) Model depicting the proposed mechanism of Akt regulation by BAG5. Under starved conditions and BAG5 overexpression, Akt interacts with BAG5, which promotes Akt inhibition and, with the participation of Hsp70, leads to its degradation. When the Akt signaling pathway is stimulated, the interaction between Akt and BAG5 decreases, coincident with Akt-dependent BAG5 phosphorylation.

## Data Availability

All data are included in the manuscript.

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
