# Peer review of "Akt Is Controlled by Bag5 through a Monoubiquitination to Polyubiquitination Switch"

_ijms, 2023, doi:10.3390/ijms242417531_

Round 1

Reviewer 1 Report

Comments and Suggestions for Authors

This manuscript by Bracho-Valdés et al., describes the interplay between BAG5 and Akt. I will keep my review brief. It is well-written throughout and shows several novel interactions and consequences. The introduction is compact. The citations feel old (only four from 2020s). However, there is nothing missing and no newer references need to be included.

I believe this manuscript should be published subject to minor changes.

537        The amino acid numbers for the “kdom” need to be shown. 150-408.

186        AAG needs to be written out in full

Figure 2D             Can the authors explain the Akt band? While HE-Akt has been depleted, there appears little effect on the second Akt. This is not discussed in the results. Or is this ERK? Representative of n=?

                                ADC already defined

Figure 3D             *** usually signifies P<0.001, change to **

Figure 4D             Please enter p value instead of n.s.

Figure 4                 Please order cancer types consistently

Figure 6                HGF concentrations need to be stated when showing a concentration gradient. Is there an explanation for pAkt not increasing as expected in panel A – BAG5? Is this representative of all experiments

556        Which DMEM – catalogue number please

601        catalogue numbers and dilutions for each antibody

615        ADC already defined

627        CHX already defined

Reviewer 2 Report

Comments and Suggestions for Authors

This study identified a novel regulatory mechanism of AKT stability.  They found that BAG5 switches monoubiquitination to polyubiquitination of AKT and increases its degradation, and this AKT-BAG5 complex is formed only under serum-starved conditions.  Overall, this manuscript has a very high quality.  Most of the results are very solid, and sufficiently support the conclusions.  Here I only have very minor concerns. 

1, The knock-down efficiency of BAG5 seems to be relatively low showing in Fig 2E.  Also, the authors need to test more than one shRNA to avoid the potential off-target effects.

2, The blot for BAG5 in Fig 6B is too weak.  Higher exposure image is required.

Comments on the Quality of English Language

The quality of language is acceptable. 
